# Exploring the Impact of Fermentation on Brown Rice: Health Benefits and Value-Added Foods—A Comprehensive Meta-Analysis

Min-Jin Lim [1,†], Kaliyan Barathikannan [1,2,†], Ye-Jin Jeong [1], Ramachandran Chelliah [1,3,4,*], Selvakumar Vijayalakshmi [1], Seon-Ju Park [5] and Deog-Hwan Oh [1,4,*]

1   Department of Food Science and Biotechnology, College of Agriculture and Life Sciences, Kangwon National University, Chuncheon 24341, Republic of Korea; mjade1@kangwon.ac.kr (M.-J.L.); barathikannan@kangwon.ac.kr (K.B.); yeajin0815@kangwon.ac.kr (Y.-J.J.); vijiselva10@kangwon.ac.kr (S.V.)
2   Agriculture and Life Science Research Institute, Kangwon National University, Chuncheon 24341, Republic of Korea
3   Saveetha School of Engineering, Saveetha (SIMATS) University, Sriperumbudur, Chennai 600124, India
4   Kangwon Institute of Inclusive Technology (KIIT), Kangwon National University, Chuncheon 24341, Republic of Korea
5   Chuncheon Center, Korea Basic Science Institute, Chuncheon, Gangwon-do 24341, Republic of Korea; sjp19@kbsi.re.kr
*   Correspondence: ramachandran865@kangwon.ac.kr (R.C.); deoghwa@kangwon.ac.kr (D.-H.O.); Tel.: +82-102-556-2544 (D.-H.O.)
†   These authors contributed equally to this work.

**Abstract:** The escalating global incidence of obesity and chronic diet-related disorders, such as type 2 diabetes, hypertension, cardiovascular disease, malignancies, and celiac disease, has intensified the focus on dietary factors and disease risks. Rice, a dietary staple for billions, is under scrutiny, particularly polished or white rice, which is high in starch and in the glycemic index and low in nutrition due to the removal of the outer bran layer during milling. This study critically analyzes the comparison between whole brown rice (BR) and milled white rice in terms of health benefits. A significant finding is the enhancement of food nutrition through fermentation, which improves protein digestibility and mineral availability and releases peptides and amino acids. The study also highlights the increased antibacterial and antioxidant activity of foods, including health benefits, through fermentation. A comprehensive review of existing data on the nutritional content and health advantages of whole fermented BR grains is presented, alongside experiments in developing fermented BR-based foods. The safety, preservation, and the economic and environmental advantages of consuming regularly fermented BR instead of white or unfermented BR are discussed. Finally, the paper addresses the commercialization challenges and future opportunities for promoting fermented BR as a healthier food alternative.

**Keywords:** brown rice; fermentation; dietary fiber; health benefits; nutritional quality; white rice





## 1. Introduction

More than half of the world's population relies on rice as their primary source of nutrition, with the vast majority preferring white rice. Brown rice, available in various colors like red, brown, black, and purple, is consumed predominantly in Asian countries [1]. Unique because of its rice bran, brown rice offers higher nutritional value than white rice, as it boasts more vitamins and dietary fiber. As noted by Pang et al. (2018), brown rice possesses significant antioxidant activity and is rich in phytochemicals. Furthermore, it contains components beneficial to health, such as gamma-aminobutyric acid (GABA), flavonoids, phenolic acid, and tocotrienol [2]. This suggests that brown rice has potential as a functional food.

Fermented foods have long been esteemed in global culinary traditions for their distinctive flavors and health benefits. The study titled 'Health-promoting fermented rice and value-added foods: a comprehensive and systematic review and meta-analysis of fermentation effects on brown rice' explored the nutritional transformation of brown rice during fermentation. Unlike white rice, brown rice retains its bran and germ, making it a nutrient-rich grain [3]. Fermentation enhances these nutritional attributes by breaking down certain anti-nutritional factors like phytic acid. This process increases nutrient bioavailability and also produces various bioactive compounds known for antioxidant, anti-inflammatory, and probiotic properties. The versatility of fermented rice has led to its inclusion in a range of products, from beverages to snacks. These not only delight the palate but also offer health benefits [4]. However, it is vital to ensure that the fermentation process remains free from harmful microorganisms.

Brown rice (BR) is well acknowledged for its significant health benefits, owing to its comprehensive nutritional profile. Structurally, the BR grain includes the pericarp (1–2%), aleurone layer and seed coat (4–6%), germ (2–3%), and a predominantly starchy endosperm (89–94%). In contrast to milled rice, which is primarily endosperm, BR boasts a rich array of bioactive compounds such as phenolic acids, flavonoids, γ-oryzanol, β-glucan, arabinoxylans, and essential vitamins like thiamine, riboflavin, and nicotinic acid. However, BR's high fiber content in its pericarp, aleurone layer, and seed coat presents challenges in industrial processing. This fiber impedes water absorption and starch expansion, leading to a prolonged cooking time and a less appealing, tougher texture.

To enhance BR's palatability while preserving its nutritional integrity, various processing methods have been explored. One such method is fermentation, a biochemical process involving microorganisms and their metabolic products (enzymes, organic acids, etc.), which enhanced the release of bioactive compounds, altered the texture, and transformed cellular components. Solid-state fermentation (SSF) is particularly advantageous due to its higher productivity, lower catabolic repression, and its ability to augment nutritional and biological properties, thus improving food quality and reducing production costs.

Specific microorganisms like *Lactobacillus plantarum* (*L. plantarum*) and *Saccharomyces cerevisiae* (*S. cerevisiae*) have long been utilized in cereal, vegetable, and fruit fermentation, enhancing texture and flavor, suppressing spoilage, and extending shelf-life in products like bread, sausages, and wine. Filamentous fungi such as *Rhizopus, Aspergillus*, and Neurospora, with species like *Rhizopus oryzae, Aspergillus oryzae*, and *Neurospora sitophila*, are notable for their enzyme production and have been recognized as safe for food use by the FDA. Fermentation with these strains significantly improves the texture, flavor, and taste of BR products. This improvement is attributed to the metabolites produced during SSF and the disruption of the grain structure, which enhanced water absorption, reduced hardness, and released volatile flavor compounds. Post-SSF treatment with strains like *L. plantarum*, BR became easier to cook, tasted better, and experienced slower starch retrogradation. Furthermore, various commercial baker's yeasts have been shown to elevate the antioxidant activities and total phenolic contents in BR.

Solid-State Fermentation (SSF) trays, which are both popular and cost-effective, are designed with metal and lids to accommodate solid substrates and inoculum. These conditions promote the uniform growth of mycelia or microorganisms. SSF supports microbial growth and the synthesis of metabolites. This method of fermentation is not only cost-effective but also environmentally friendly, especially when compared to enzymatic methods in the bioactive compound synthesis industry [5].

Although SSF's role in enhancing BR's physicochemical and functional qualities has been well researched, its impact on BR's bioactive components and the structural changes in its starch and fiber remain underexplored. This study focuses on treating BR with SSF using *L. plantarum*, *S. cerevisiae*, *R. oryzae*, *A. oryzae*, and *N. sitophila* to analyze its impact on the content of bioactive compounds and the grain's microstructure. Key components like β-glucan, arabinoxylan, γ-oryzanol, thiamine, riboflavin, nicotinic acid, and polyphenols are measured. The study also employs scanning electron microscopy (SEM) to compare

the microstructure of BR grains before and after SSF treatment. The primary goal is to identify the optimal SSF strain for enhancing processing quality, focusing on the content and bioavailability of major bioactive compounds, to facilitate the production of BR-based foods and boost their consumer appeal.

Ferulic acid (FA) is a prominent phenolic substance found in brown rice bran. Fermented brown rice contains ferulic acid and other physiologically beneficial components like oryzanol, known for their cholesterol-lowering effects [6]. According to Vinayagam et al., dietary FA has been shown to positively affect intestinal lipid excretion, lipogenic enzyme activity, and glucose metabolism in HFD rats, thereby reducing hyperglycemia [7]. Fermentation, a process involving the conversion of carbohydrates to alcohol or organic acids using microorganisms—yeasts or bacteria—under anaerobic conditions, is a traditional method of food preservation that also enhances food properties. When brown rice undergoes fermentation, particularly with the addition of probiotics (live beneficial bacteria), this process yields a range of bioactive chemicals. These chemicals include various types of organic acids, enzymes, and bioactive peptides, which are known for their health-enhancing properties. Brown rice, inherently nutritious due to its high fiber content, essential fatty acids, vitamins, and minerals, becomes even more beneficial when fermented. Fermentation breaks down complex carbohydrates and proteins in brown rice, making it easier to digest and allowing the body to absorb nutrients more effectively. Additionally, the presence of probiotics in fermented brown rice contributes to gut health by enhancing the gut microbiota, which plays a crucial role in digestion and overall health.

Research has indicated that fermentation augmented the nutritional and functional qualities of grains. For instance, fermented brown rice has displayed considerable antioxidant capacity. The fermentation process increased the presence of total phenols, flavonoids, and anthocyanins. The application of fermentation across various studies has consistently improved food quality and created nutritionally and functionally superior products. It has genuinely added value [1]. Our review examined the constituents of brown rice and their potential applications in producing functional meals. This presented the opportunity to produce an array of products from fermented brown rice, catering to consumer demands in the food industry and offering significant health advantages.

## 2. Nutritional Quality of Brown Rice and Its Significance in Enhancement Using Bioconversion Technology

Brown rice, a less processed form of rice, retains its bran and germ layers. This preserves a plethora of vital nutrients that are lost during the refining process of white rice. With the growing awareness of the health benefits associated with whole grains, the nutritional quality of brown rice has garnered significant attention from both consumers and researchers. Moreover, advancements in bioconversion technology offer a promising avenue to further enhance the nutritional profile of this staple grain. Brown rice is an excellent source of macronutrients such as carbohydrates, which provide energy. It also boasts a higher protein content compared to white rice [8]. The bran layer contributes dietary fiber, which aids in digestion and promotes satiety. Furthermore, brown rice is rich in essential micronutrients like magnesium, phosphorus, selenium, and zinc. It contains vitamins such as niacin (B3), pyridoxine (B6), and folate. Additionally, it includes phytochemicals and a variety of bioactive compounds like phenolic compounds, flavonoids, and γ-oryzanol, which exhibit antioxidant properties. However, it also has a high level of phytic acid, which can act both as an antioxidant and as a potential inhibitor of nutrient absorption. Brown rice also contains lipids with essential fatty acids, particularly in the bran oil [3].

Bioconversion, which entails the use of biological systems like microorganisms and enzymes, transforms one material into another, often amplifying its nutritional or functional properties. For instance, phytic acid in brown rice can inhibit the absorption of certain minerals [3]. Bioconversion methods, especially fermentation, can reduce its phytic acid content, thereby enhancing mineral bioavailability. Some microbial processes break down proteins to release amino acids or enhance the amino acid profile of rice by introducing

essential amino acids that might have been in short supply [9]. Fermentation, a subset of bioconversion, spurs the production of certain vitamins [10]. For example, specific microorganisms generate B-vitamins during fermentation. Microbial fermentation also leads to the formation of new bioactive compounds that offer health benefits, such as peptides with antioxidant or antihypertensive properties [5].

During fermentation, different varieties of brown rice were examined. The process revealed variations in the nutrient content of fermented brown rice. Fermentation increases the levels of some nutrients like zinc, phosphorus, magnesium, iron, and calcium, while reducing others. For instance, the phytic acid content decreases by approximately 41%, a reduction deemed highly effective as cited in the Food Chemistry journal [3]. Solid-fermented brown rice exhibits a similar decrease in phytic acid [11]. Additionally, while brown rice is a recognized source of oryzanol and the vitamin E isoform, fermented brown rice shows a significant increase in levels of -tocotrienols, with a marked decrease in levels of -tocopherols [3]. Research by Ma et al. in 2023 indicated that the outcomes for fermented brown rice are 1.72% superior to those of its unfermented counterpart [12]. GABA, or gamma-aminobutyric acid, was found in greater concentrations in fermented brown rice than in unfermented brown rice. A study by Tyagi et al. in 2021 underscored that one of the nutritional boons of the fermentation process was the increased presence of GABA [8].

Compared to white rice, which undergoes a process that removes many essential parts, brown rice retains most of its nutrients. Brown rice's nutritional quality and its enhancement through bioconversion technology deserves discussion. Being nutritionally dense, brown rice offers a myriad of macronutrients, micronutrients, and other beneficial compounds (Table 1). The potential of bioconversion technology, especially fermentation, further elevates these benefits (Figure 1). As the demand for nutrient-enriched foods escalates, melding traditional whole grains like brown rice with innovative bioconversion methods holds great promise for the future of food science and nutrition.

**Table 1.** A comprehensive comparison detailing the nutritional composition, benefits, and drawbacks of both brown and white rice. This analysis is based on extensive data sourced from references [2,3,9,10,13]. The table methodically enumerates key nutritional parameters such as calorie content, macronutrient ratios, vitamins, and mineral content, providing a clear distinction between these two varieties of rice. Additionally, it explores the health implications of each type, including their impact on dietary considerations like their fiber content, and their overall contribution to a balanced diet.

| | Brown Rice (*Oryza sativa*) | | | | | | | | | | |
|---|---|---|---|---|---|---|---|---|---|---|---|
| Variety | Bapata | Calrose | X2 | T15 | Q34 | Red Rice | Black Rice | Chak-Hao Angangba | Chak-Hao Poireiton | Chak-Hao Amubi | Uma |
| Protein (g/100 g) | 7.8 | 6.87 | 8.92 | 9.57 | | 8.1 | 8 | 5.57 | 7.77 | 8.75 | 7.6 |
| Carbohydrate (g/100 g) | 71.3 | | | | | 85.1 | 87.6 | 78.24 | 74.38 | 74.67 | 72.2 |
| Fat (g/100 g) | 2.4 | 2.43 | 2.43 | 2.86 | | 1.1 | 2.9 | 3.05 | 3.73 | 3.33 | 2.3 |
| Ash (g/100 g) | 1.1 | 1.24 | 1.67 | 1.35 | | 1.7 | 1.5 | 1.38 | 1.79 | 0.83 | 1.0 |

**Table 1.** *Cont.*

| | Brown Rice (*Oryza sativa*) | | | | | | | | | | |
|---|---|---|---|---|---|---|---|---|---|---|---|
| **Variety** | **Bapata** | **Calrose** | **X2** | **T15** | **Q34** | **Red Rice** | **Black Rice** | **Chak-Hao Angangba** | **Chak-Hao Poireiton** | **Chak-Hao Amubi** | **Uma** |
| Calcium (mg/kg) | | | 148.85 | 160.71 | 490.00 | | | 77.6 | 114.6 | 136.2 | |
| Magnesium (mg/kg) | | | 437.42 | 24.59 | | | | 379.1 | 387.6 | 377.2 | |
| Phosphorus (mg/kg) | | | | | 2010.00 | | | 2248.1 | 2529.7 | 2062.1 | |
| Potassium (mg/kg) | | | | | 585.00 | | | 1546.8 | 1843.6 | 1606.6 | |
| Copper (mg/kg) | | | 1.88 | 1.58 | | | | 30.6 | 27.5 | 33.4 | |
| Manganese (mg/kg) | | | 439.03 | 22.05 | | | | 23.6 | 42.7 | 38.8 | |
| Iron (mg/kg) | | | 22.66 | 19.64 | 60.05 | | | 57.1 | 47.2 | 88.8 | |
| Zinc (mg/kg) | | | 16.09 | 22.05 | | | | 34.9 | 42.4 | 53.9 | |

| Advantages of Brown Rice (BR) and White Rice (WR) Intake | <ul><li>BR—decreases body fat, reduces the total cholesterol and triglycerides</li><li>BR—decreases fat accumulation by adipocyte differentiation inhibition.</li><li>BR—protects against apoptosis</li><li>BR—reduces plasma glucose</li><li>BR—reduces postprandial glucose level</li><li>BR—ameliorates cardiovascular disease risk by modulating lipid metabolism and oxidative stress</li><li>BR—antiproliferative action</li><li>BR—reduces glucose excursions</li><li>BR—decreases the markers of liver inflammation and fibrosis</li><li>WR—white rice is an excellent source of manganese, providing over 30% of the daily value (DV). It is also a good source of iron, providing 2.7 mg or 15% of the DV. White rice also supplies B vitamins (especially thiamin, but also niacin and riboflavin).</li></ul> |
|---|---|
| Disadvantage of Brown Rice Intake and Overintake of White Rice | <ul><li>BR—brown rice tends to have a much higher arsenic content than white rice. Arsenic is a toxic element that accumulates in the outer layers (germ and bran) of brown rice. These layers are removed in the process of making white rice, which contains about 80% lower arsenic than the same variety of brown rice</li><li>BR—brown rice is full of phytates and lectins, which bind to vitamins and minerals and decrease the bioavailability.</li><li>WR—white rice is highly processed and missing its hull (the hard protective coating), bran (outer layer) and germ (nutrient-rich core)</li><li>WR—higher intake of white rice leads to high blood pressure, higher fasting blood sugar (higher glycemic index), higher triglyceride levels and low levels of HDL cholesterol, and Brown rice vice versa</li></ul> |

**Table 1.** *Cont.*

| | Polished rice (*Oriza sativa* L.) | | | | | | | | | | |
|---|---|---|---|---|---|---|---|---|---|---|---|
| Variety | Bapata | Calrose | X2 | T15 | Q34 | Red rice | Black rice | Chak-hao Angangba | Chak-hao Poireiton | Chak-hao Amubi | Uma |
| Protein (g/100 g) | 6.3 | 6.01 | 8.10 | 8.68 | | 7.6 | 6.7 | 5.29 | 7.45 | 8.48 | 6.4 |
| Carbohydrate (g/100 g) | 80.1 | | | | | 91.1 | 92.4 | 85.57 | 83.27 | 82.13 | 78.0 |
| Fat (g/100 g) | 0.7 | 0.86 | 1.16 | 1.15 | | 0.7 | 0.4 | 0.21 | 0.34 | 0.41 | 1.2 |
| Ash (g/100 g) | 0.5 | 0.35 | 0.70 | 0.48 | | 0.6 | 0.5 | 0.31 | 0.57 | 0.33 | 0.6 |
| Calcium (mg/kg) | | | 105.29 | 126.78 | 109.00 | | | 42.5 | 63.2 | 53.6 | |
| Magnesium (mg/kg) | | | 188.64 | 11.53 | | | | 56.8 | 215.8 | 106.6 | |
| Phosphorus (mg/kg) | | | | | 1085.00 | | | 456.7 | 1401.6 | 718.5 | |
| Potassium (mg/kg) | | | | | 350.00 | | | 449.2 | 966.4 | 566.1 | |
| Copper (mg/kg) | | | 1.42 | 1.06 | | | | 24.1 | 20.1 | 26.2 | |
| Manganese (mg/kg) | | | 180.20 | 15.26 | | | | 14.5 | 21.8 | 20.1 | |
| Iron (mg/kg) | | | 112.96 | 9.73 | 16.00 | | | 24.5 | 30.6 | 26.2 | |
| Zinc (mg/kg) | | | 14.52 | 15.26 | | | | 20.2 | 24.7 | 24.6 | |

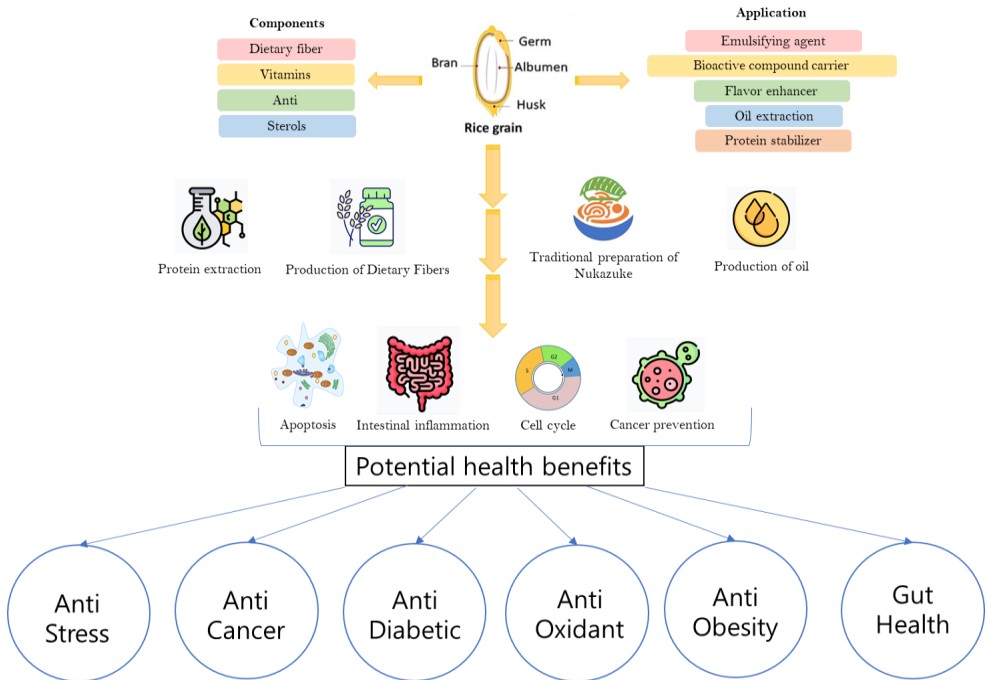

**Figure 1.** Illustration of the diverse applications and nutritional benefits of brown rice in a multi-dimensional context. This visual representation delves into the various aspects of brown rice, highlighting its versatility in different culinary uses and its role in various dietary regimes. The figure also comprehensively outlines the nutritional profile of brown rice, including its rich content of essential nutrients like fiber, vitamins, and minerals. Special emphasis is placed on the health benefits associated with its regular consumption, such as improved digestion, better glycemic control, and contribution to a heart-healthy diet. The different color squares were provided for the different compounds because of different categories and the application also widely applied. In addition, this figure serves as an informative guide, showcasing the wide-ranging advantages of incorporating brown rice into daily meals.

## 3. Potential Health Benefits of Bioactive Compounds on Human Health

Globally, rice is one of the most widely consumed staple foods and plays a pivotal role in human nutrition. Brown rice, unlike its refined counterpart, white rice, retains its bran and germ. These parts are rich in bioactive compounds such as phenolic compounds, which counteract free radicals and possess potential anti-diabetic, anti-obesity, anti-inflammatory, and anticancer properties [13–16] Figure 2. Ferulic acid, a significant phenolic acid in brown rice's bran, is renowned for its antioxidant capacity. Flavonoids, known for their antioxidant, anti-inflammatory, and potential anticancer properties, are also present. Unique to rice bran oil, γ-oryzanol is researched for its antioxidant and cholesterol-lowering capabilities [5]. The phytic acid in brown rice, which may inhibit the absorption of some minerals, also acts as an antioxidant that could protect against certain diseases. Moreover, brown rice bran contains inositol-hexaphosphate (IP6) compounds, recognized for their potential anticancer activity [17].

The majority of the bioactive compounds in brown rice function as antioxidants, neutralizing free radicals and possibly decreasing the risk of diseases linked with oxidative stress, such as cardiovascular diseases [14]. The anti-inflammatory properties of bioactive compounds in brown rice, especially flavonoids, can modulate inflammatory pathways, reducing the risk of inflammation-driven diseases. γ-Oryzanol, a primary bioactive compound found mainly in brown rice bran oil, can significantly reduce LDL (bad cholesterol) levels, potentially benefiting cardiovascular health. It also acts as a hormonal modulator influencing endocrine function, provides relief from menopausal symptoms, and has neuroprotective qualities due to its antioxidant properties [5]. There is also substantial evidence

supporting γ-Oryzanol's role in muscle building, increasing testosterone, and enhancing athletic performance.

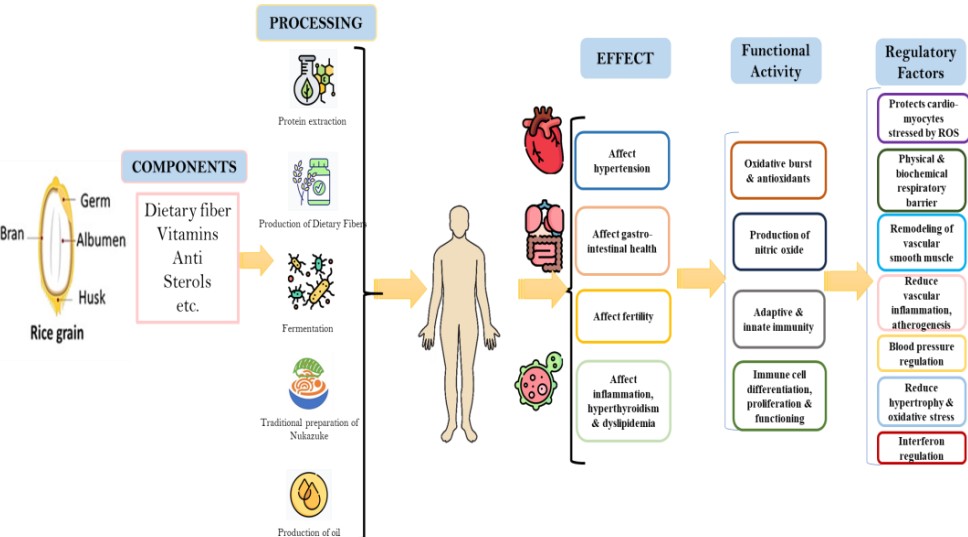

**Figure 2.** Diagrammatic representation that focuses on the effects of brown rice on human health, particularly emphasizing its immunomodulatory efficacy. This detailed illustration captures the multifaceted impact of brown rice consumption on the human immune system. It outlines how various components of brown rice, such as antioxidants, vitamins, and minerals, contribute to enhancing the immune response and regulating immune functions. The diagram also depicts the pathways through which brown rice influences overall health, including its role in reducing inflammation, combating oxidative stress, and supporting gut health. This visual aid is designed to convey complex scientific information in an accessible manner, highlighting the significant health benefits of brown rice in boosting and modulating the immune system.

Several compounds, notably IP6 and phenolic compounds, have shown potential anticancer activities by inducing apoptosis (programmed cell death) in cancer cells and inhibiting tumor growth [17]. Additionally, the consistent consumption of whole grains like brown rice is linked to a reduced risk of type 2 diabetes. Bioactive compounds can improve insulin sensitivity and decrease postprandial glucose levels. The significant dietary fiber content in brown rice, along with certain bioactive compounds, can promote gut health by fostering a healthy microbiome and ensuring regular bowel movements [14].

These compounds are increasingly acknowledged for their potential health benefits, especially in disease prevention and health promotion contexts. Researchers have extensively studied the numerous health benefits of brown rice over the years. Fermented brown rice, for instance, lowers cholesterol levels and protects the liver from free radical damage due to copper accumulation [15]. Moreover, fermented brown rice exhibits anti-colitis, anti-cancer, prebiotic, chemo-preventive, and anti-inflammatory properties, highlighting its potential as a functional food for healthcare [18]. Another study found that fermented red brown rice protected DNA from oxidative stress-induced damage [19]. Additionally, fermented brown rice has a more palatable texture than its non-fermented counterpart, and extrusion processes can further improve brown rice's natural elasticity and water absorption rate [12]. GABA, a central nervous system neurotransmitter, performs various physiological functions, including preventing cardiovascular disease and regulating blood pressure, hormones, and cells. *L. reuteri* (AKT1) displayed the highest GABA concentration among ten different Lactobacillus strains detected by HPLC [13]. Thus, the bioactive compounds in brown rice offer a multitude of potential health benefits, from antioxidant and anti-inflammatory effects to cholesterol regulation and cancer prevention [15]. These discoveries highlight the importance of incorporating brown rice into a healthy diet and

underscore the need for further research to thoroughly understand the mechanisms through which these bioactive compounds deliver their benefits.

### 3.1. Gamma-Aminobutyric Acid (GABA) Levels in Brown Rice

GABA, crucial for its inhibitory role in the nervous system and influence on various physiological and mental health aspects, was measured in brown rice samples subjected to different treatments. Notably, *Lactobacillus reuterii* (*L. reuterii*) emerged as the most effective strain among 10 different lactic acid bacteria strain (*P. pentosaceus (FMC1)*, *L. fermentum (FMF2)*, *L. fermentum (AKT2)*, *L. rhamnosus (FMR1)*, *L. rhamnosus (FMR2)*, *L. brevis (FMB1)*, *L. brevis ATCC (STANDARD)*, *L. plantarum (FMP1)*, *L. plantarum (FMP2)* and *L. reuterii (AKT1)*) varieties for enhancing GABA content during fermentation, a finding backed by rigorous statistical analysis. The increase in GABA concentration from a mere 1.61 µg/mL in raw rice to 27.03 µg/mL in *L. reuterii*-fermented rice marks a significant enhancement, surpassing previous research benchmarks [1,2]. This study did not just stop at fermentation; it delved deeper by comparing fermentation-only and germination-only processes, as well as their combination. Interestingly, germination alone led to the highest GABA levels, with the 48 h germination period being particularly effective. This gave a valuable perspective for optimizing GABA production in brown rice, an aspect crucial for leveraging its health benefits. In summary, this review contributes significantly to the understanding of how brown rice processing can be optimized for maximum GABA production, providing essential insights for both food science and nutritional health [3].

### 3.2. Amino Acid Composition in Brown Rice

Amino acids, crucial for organism development and flavor enhancement in foods, were thoroughly analyzed across different brown rice samples. The review employed meticulous analytical methods, including ANOVA and PCA, revealing that raw brown rice possesses the lowest amino acid content. In contrast, fermentation, especially with *L. reuterii*, markedly increased amino acid levels. This enrichment is attributed to enzymatic activities during fermentation that break down the food matrix, releasing various beneficial metabolites and bioactive compounds. Germination also played a notable role in augmenting the amino acid content, activating dormant enzymes, and thus enriching the rice's nutritional profile [1]. Fascinatingly, fermentation led to a significant rise in several essential amino acids, with a notable increase in some conditionally essential amino acids (Figure 3a) [8]. The combined germination and fermentation approach further amplified these effects, as demonstrated through comprehensive PCA analysis. The dynamic changes in brown rice's nutritional profile through different processing methods also underscore the importance of such treatments in enhancing the health benefits of this staple grain [6].

### 3.3. Total Phenolic Content (TPC) and Total Flavonoid Content (TFC) in Brown Rice

Phenolics, characterized by their hydroxyl groups on aromatic rings, are known for their antioxidative effects. A previously reported study meticulously quantified TPC and TFC in various brown rice samples, with *L. reuterii*-fermented rice showing the highest TPC, a significant increase from the levels in raw rice. This elevation in phenolic content, observed both in germinated and fermented samples, was credited to enzymatic hydrolysis, underscoring the effectiveness of these processing methods in enhancing the rice's health benefits [1,8]. Investigating the individual phenolic compounds in brown rice using UPLC-ESI-Q-TOF-MS/MS technology led to the identification of fourteen phenolic compounds (Figure 3b), known for their pivotal role in combating chronic diseases. The study adeptly shows the variations in phenolic compound concentrations across different brown rice processing methods. Notably, fermentation, especially with *L. reuterii*, significantly elevated these beneficial compounds, surpassing those found in germinated or raw rice [8]. This enhancement is likely due to fermentation's ability to release bound phenolic compounds, improving their bioavailability in the grain. The reports are widely depicted in a heat map analysis, offering a clear, visual representation of the concentration gradients. Among

the detected antioxidants, compounds like β-carotenol and eugenol stood out, recognized for their strong antioxidant capacity and stress-reducing effects [4]. The use of PCA further corroborated these results, distinguishing the fermented brown rice as a superior source of phenolic compounds compared to other samples. This comprehensive analysis not only underscores the health benefits of fermented brown rice but also highlights the transformative impact of processing methods on the nutritional profile of this staple grain.

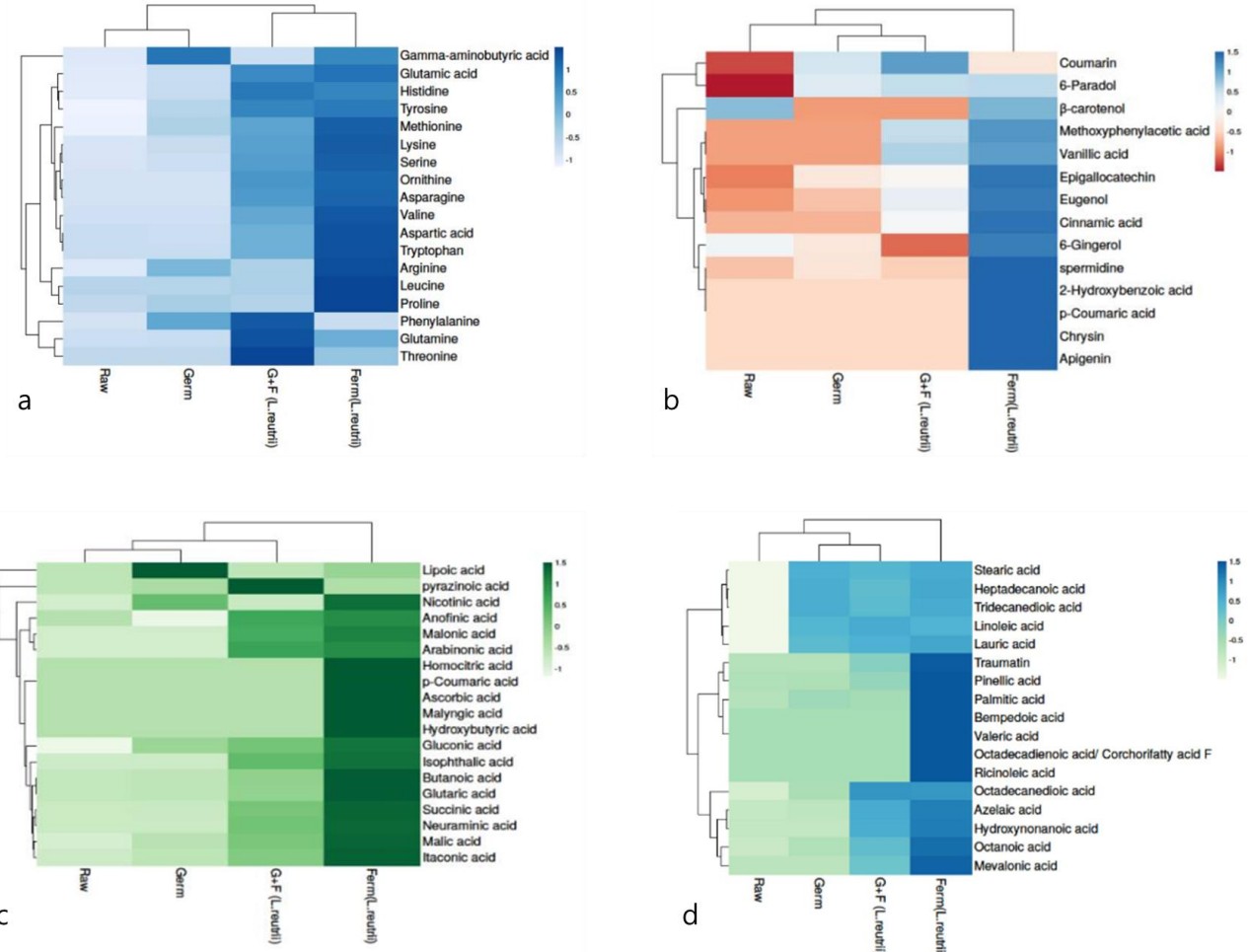

**Figure 3.** (**a**) Represents a comprehensive analysis of amino acid levels in various rice samples: raw, germinated, fermented with *L. reuteri*, and a combination of germination and fermentation (G+F) with *L. reuteri*. The figure uses a heat map with varying shades of blue to illustrate the differences in amino acid concentrations among these samples, providing a clear visual comparison; in (**b**), the focus shifts to the levels of phenolic compounds in the same types of rice samples. This section employs a heat map that transitions from blue to red, with the intensity of the color indicating the concentration of phenolic compounds. The color gradient allows for an immediate visual interpretation of the phenolic content, with red representing higher levels and blue indicating lower levels; (**c**) explores the levels of organic acids in the raw, germinated, fermented (*L. reuteri*), and G+F (*L. reuteri*) rice samples. Here, the heat map employs a color spectrum from green to white, where green signifies higher concentrations of organic acids, and white indicates lower concentrations. This visual tool effectively demonstrates the variation in organic acid content across different rice processing methods. (**d**) examines the fatty acid levels in these rice samples. The heat map for this section uses a color range from blue to green, where blue represents higher fatty acid levels and green shows lower levels. This visual representation provides an intuitive understanding of how fatty acid content varies between the raw, germinated, fermented, and G+F samples (Source of the figure: reference [1,8]).

Likewise, flavonoids, forming a significant portion of these phenolic compounds, also displayed higher concentrations in fermented rice, again highlighting the positive impact of fermentation on the rice's antioxidative properties [4]. This study's findings on TFC and TPC surpass some previous research studies while falling short of others, suggesting the influence of various factors like rice genotype and environmental conditions on these components [5,6]. They showed the complexities of phenolic and flavonoid content measurement, noting that extraction methods can significantly influence results. Overall, the review provides a valuable contribution to understanding how different processing methods can optimize the health-promoting properties of brown rice, a staple food with profound global significance.

### 3.4. Organic Acid Levels in Brown Rice

Organic acids have given significant insights into how different processing methods like germination and fermentation affect their concentration. The identification of nineteen organic acids (Figure 3c), including notable antioxidants such as ascorbic acid and p-coumaric acid, underscores the potential health benefits of these rice treatments. The statistical analysis, represented through a heat map and confirmed by PCA, revealed notable differences in organic acid content among the variously processed rice samples. Fermentation, particularly with *L. reuterii*, emerged as a highly effective method for enhancing organic acid levels, attributed to the microbial breakdown of the cell wall and subsequent release of bioactive compounds [1]. Similarly, germination was shown to play a significant role in increasing these beneficial compounds, due to cellular respiration and molecule degradation processes [3]. The PCA analysis provided an in-depth understanding of these variations, distinguishing the unique profile of fermented brown rice from other samples. In essence, this research not only highlights the superior organic acid profile of fermented brown rice but also demonstrates the transformative effects of germination and fermentation on the nutritional and functional qualities of this staple grain [8]. The findings offer valuable guidance for optimizing brown rice processing to maximize its health-promoting properties.

### 3.5. Fatty Acid Levels in Brown Rice

The fatty acid composition of brown rice offers critical insights into the impact of various processing methods on these essential nutrients. The detection of seventeen distinct fatty acids across different brown rice samples underscores the significant influence of treatment methods on their concentrations [9]. The highest fatty acid levels (Figure 3d) were observed in fermented brown rice, affirming the role of fermentation not just in enhancing bioactive compound bioavailability but also in generating beneficial end-products like short-chain fatty acids (SCFAs), known for their health advantages. Utilizing heat map analysis for an intuitive representation of fatty acid concentrations, the study identified fatty acids like stearic and linoleic acid, which are celebrated for their antioxidative and stress-reducing properties [1,8]. The use of Principal Component Analysis (PCA) further deepens the understanding of these variations, revealing a unique fatty acid profile in fermented brown rice, particularly those treated with *L. reuterii*. Additionally, a notable correlation between germinated and combined germination and fermentation (G+F) samples was observed [8]. Overall, this review provides a valuable perspective on the nutritional enhancement of brown rice through fermentation and germination, contributing significantly to the development of healthier rice-based food products.

### 3.6. Bioactive Peptides Identified in Brown Rice Samples

This insightful study sheds light on the efficacy of microbial fermentation as a method for deriving bioactive peptides from brown rice. Highlighting microbial fermentation as not only a cost-effective alternative to enzyme uses but also a more comprehensive approach, the study emphasizes its ability to break down proteins and mitigate anti-nutritional factors [8]. The research predominantly found peptides in fermented brown rice

samples, specifically those treated with *L. reuterii*, suggesting a rich potential for producing beneficial compounds through fermentation [10]. A detailed analysis of these peptides, comparing them with known peptides in the literature, revealed a total of 10 identifiable peptides with potential health benefits [8]. However, the study also faces the challenge of unknown peptides, which are not yet documented in existing databases, posing difficulties in understanding their full range of functions. Nonetheless, the identified bioactive peptides in fermented samples are recognized for their strong antioxidative and stress-reducing properties [12]. This review reports underscores the need for further functional analysis to fully ascertain the health benefits of these peptides, particularly in fermented brown rice. In addition, the nutritional enhancement of brown rice through fermentation provides a promising avenue for developing health-oriented rice products.

## 4. Enhanced Antioxidant Efficacy of Processed Brown Rice and Its Potential Applications

Humans rely on the inhalation of oxygen for essential energy production and bodily functions. However, excessive oxygen generates reactive oxygen species (ROS), which are excessive oxygen molecules that harm healthy body cells. Such oxidative stress contributes to aging and the onset of various diseases [8]. The activation of free radicals in the body adversely affects cell structure, tissue, and DNA. This results in inflammation, apoptosis, or abnormal cell proliferation, potentially leading to tumor development. Oxidative stress arises from an imbalance between the production and removal of free radicals [9]. To counteract this, antioxidants serve a crucial role, protecting the body from oxidative stress by neutralizing endogenously produced ROS. Recently, the fermentation process was identified as a method to enhance the antioxidant activity of grains [10]. To evaluate the antioxidant potential of fermented brown rice, tests such as DPPH (2,2′-diphenyl-1-picrylhydrazyl) radical scavenging activity using spectrophotometry, ABTS radical scavenging activity, and FRAP analysis were employed. The results showed that fermented brown rice surpassed unfermented brown rice in all three measures [8,13]. Among unfermented raw brown rice, germinated brown rice, fermented brown rice, and mixtures of the two, fermented brown rice exhibited the highest DPPH and ABTS radical-scavenging activities [8]. This superior antioxidant capacity was also corroborated using FRAP analysis. Furthermore, *L. reuteri* FBR recorded the highest activity, followed closely by *L. plantarum* and *L. fermentum* FBR.

Peptides produced by *L. reuteri*-fermented brown rice are particularly noteworthy. Out of these, peptides containing aromatic and hydrophobic amino acids were selected due to their potential to enhance lipid interactions or serve as vital proton/hydrogen donors. The antioxidant activities of these selected peptides, as gauged by DPPH, ABTS, and FRAP analyses, were found to be robust and compelling [20]. This deep dive into the antioxidant properties of fermented brown rice emphasizes its significant potential in combating oxidative stress, underscoring its value in nutrition and health (Figure 4).

The enhanced antioxidant potential achieved through fermentation, particularly with *L. reuteri*, led the charts in the DPPH assay for radical scavenging activity [6,8]. This trend of superior performance by fermented rice was consistently observed across all assays, including ABTS and FRAP, positioning it above both germinated and raw brown rice in antioxidant capacity. The previous reports have aligned with the principle that fermentation and germination processes elevated the healthful attributes of cereals, a fact reflected in the significantly higher antioxidant activities in treated brown rice samples compared to raw ones [12]. Overall, this review provided valuable insights into optimizing brown rice processing for enhanced health benefits, particularly in terms of bolstering its antioxidant properties.

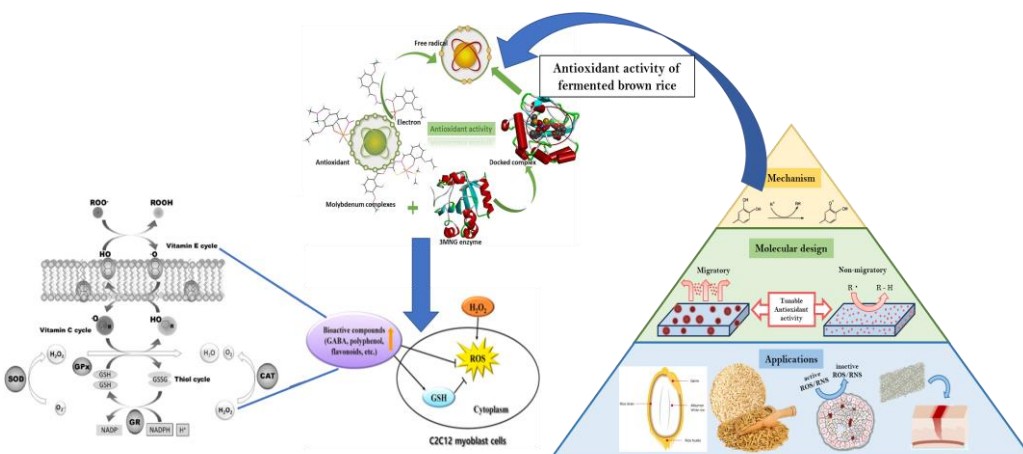

**Figure 4.** Visual representation of the role of the antioxidant mechanism and its application in microbial fermented brown rice. This figure is designed to elucidate the complex interactions and benefits of antioxidants in fermented brown rice, emphasizing how fermentation enhances the bioavailability and efficacy of these antioxidants. It highlights the specific types of antioxidants present in brown rice and how microbial fermentation, particularly with beneficial bacteria, can increase their potency. The figure also showcases various applications of this process, illustrating how enhanced antioxidant activity in fermented brown rice can contribute to improved health outcomes, such as better oxidative stress management, increased nutritional value, and potential therapeutic uses. Additionally, the diagram indicates the biochemical pathways involved in antioxidant activity, providing a comprehensive view of the process from fermentation to the enhanced health benefits.

## 5. Processed Brown Rice Plays a Significant Role in Exhibiting Anti-Diabetic Properties

Diabetes is a condition where glucose from the blood enters the cells but cannot not be utilized for energy, leading to elevated blood sugar levels. There are various types of diabetes: type 1, type 2, gestational diabetes, and others. In type 1 diabetes, insulin production is halted due to the disease destroying the beta cells in the pancreas. In type 2 diabetes, the body develops increased insulin resistance [9]. As a result, there is a marked reduction in insulin secretion, which hinders the action of insulin and elevates blood sugar levels. Diabetes remains one of the leading causes of death worldwide, with its prevalence rising steadily each year [10].

In 2022, studies indicated that functional foods are rich in anti-diabetic ingredients. Brown rice, as a whole grain, consists of the bran, germ, and endosperm. Numerous reports have suggested that its composition of complex carbohydrates, dietary fiber, and essential nutrients, such as magnesium, reduces the risk of type 2 diabetes [21–23]. These components primarily regulate the post-meal glucose response and delay carbohydrate digestion and absorption, leading to a gradual increase in blood sugar levels post-consumption [9]. Additionally, magnesium in brown rice is crucial for carbohydrate metabolism. A deficiency in magnesium can disrupt insulin secretion from the pancreas and augment insulin resistance. Moreover, phenolic compounds in brown rice, which exhibit antioxidant activities, are considered beneficial, especially since oxidative stress is linked to diabetes onset [22].

Fermented brown rice was found to have superior anti-diabetic activity compared to its unfermented counterpart. This was attributed to the enhancement of specific bioactive compounds during the fermentation process, which results in the production of compounds like peptides and γ-aminobutyric acid (GABA) [8]. GABA, for example, is known for its potential anti-diabetic effects. Furthermore, fermentation in brown rice alters its dietary fiber, potentially improving its beneficial impact on post-meal glucose responses [8–23]. This process also reduces anti-nutritional factors, such as phytic acid, which hinders mineral absorption but is degraded during fermentation. Although no direct link between phytic acid and diabetes has been established via the NF-κB pathway mechanism, enhanced

mineral bioavailability could support metabolic health [4]. Additionally, the fermentation of dietary fibers by gut bacteria produces SCFAs, known to benefit glucose metabolism.

In one study, brown rice fermented with *Aspergillus oryzae* (referred to as FBRA) was tested on female mice, illustrating its potential to suppress type 1 diabetes [14]. It was determined that consuming fermented *oryzae* and rice bran treated with *Aspergillus oryzae* could help prevent type 1 diabetes. According to a 2021 study by Kataoka et al., mice on a diet supplemented with 0.5% FBRA had a decreased risk of type 1 diabetes and insulitis [14]. Both fermented and unfermented brown rice products exhibit potential anti-diabetic benefits due to their nutrient composition and the compounds formed during fermentation. However, while promising, it is vital to integrate these potential benefits into a comprehensive balanced diet and lifestyle for diabetes management [14]. More clinical trials and research are essential to corroborate these findings and decipher the exact mechanisms at play.

## 6. Anti-Obesity and Cholesterol-Lowering Activity of Processed Brown Rice

Processed brown rice and its constituents have garnered interest in the fields of nutrition and health because of their potential anti-obesity and cholesterol-lowering properties. Conditions like diabetes and hyperlipidemia often accompany obesity and can lead to various health consequences [17,21]. Specifically, obesity is characterized by an abnormally high accumulation of fat in the body's adipose tissue. At current, over one billion people worldwide are obese, including 650 million adults, 340 million adolescents, and 39 million children. This number continues to rise rapidly. The World Health Organization (WHO) projected that by 2025, approximately 167 million adults and children will be overweight or obese, negatively impacting their health [5]. One method to combat obesity is consuming low-calorie foods. While increased physical activity is another recommended strategy, for those who find it challenging, opting for low-calorie meals is a viable alternative. Numerous natural compounds derived from plants and grains, especially fermented brown rice, have demonstrated efficacy against obesity [5]. For instance, fermented brown rice exhibits a higher lipase inhibitory activity than its unfermented counterpart. In a *Caenorhabditis elegans* model, those on a fermented-brown-rice diet had an extended lifespan and reduced cholesterol levels. Furthermore, research indicated that fermented brown rice actively prevented obesity development when included in the diet [15].

The anti-obesity activity of processed brown rice could be attributed to various factors (Figure 5). Dietary fiber, which is more abundant in brown rice compared to white rice, can enhance satiety, potentially reducing food intake [15]. This fiber could also reduce the absorption of some dietary fats and influence the release of hunger-regulating hormones. When processed brown rice underwent specific cooking and cooling methods, an increase in resistant starch was observed. This starch type was not easily digested in the small intestine and got fermented in the large intestine, which was associated with benefits like improved insulin sensitivity and potential anti-obesity effects [13,22]. As for the cholesterol-lowering properties of processed brown rice, there was an observed increase in phytosterols and saponins, both known to aid in reducing cholesterol [7]. Phytosterols hinder cholesterol absorption in the intestines, and saponins could bind to cholesterol, preventing its reabsorption. Moreover, the elevated levels of ferulic acid in fermented brown rice bran, known for its antioxidant properties, positively influence cholesterol metabolism [7]. Additionally, γ-Oryzanol reduces overall cholesterol levels, and dietary fiber in brown rice binds to cholesterol, decreasing its absorption in the gut. The potential health advantages of brown rice depend on its processing methods and refining degree [13]. Minimal processing, which retains most of the bran and germ, is generally perceived as beneficial. However, individual responses can differ. It is essential to view these health benefits as components of a well-rounded diet and healthy lifestyle. For significant dietary changes, consultation with a healthcare or nutrition professional is always recommended.

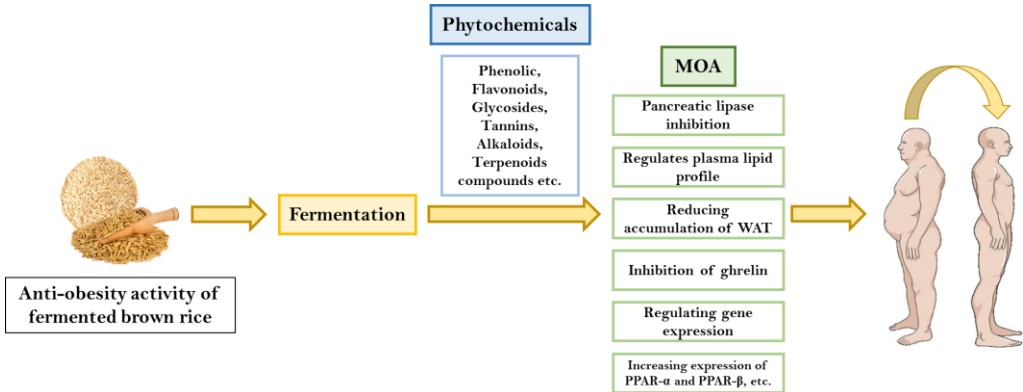

**Figure 5.** Schematic representation that details the phytochemical composition of processed (fermented) brown rice and its anti-obesity effects. This diagram focuses on identifying and illustrating the various phytochemicals that are either inherent in brown rice or enhanced through the fermentation process. It elaborates on how these phytochemicals contribute to combating obesity. The representation outlines the metabolic pathways influenced by these compounds, including their roles in fat metabolism, appetite regulation, and insulin sensitivity. The different color squares were provided for the different phytochemicals because of different categories and the application also widely applied. Additionally, the figure illustrates the specific anti-obesity mechanisms, such as the reduction in adipose tissue, inhibition of lipid accumulation, and enhancement of energy expenditure. This schematic aims to visually convey the complex interactions between the phytochemicals in fermented brown rice and their potential health benefits, particularly in the context of obesity prevention and management.

## 7. Significant Comparison of the Anti-Cancer and Anti-Inflammatory Activities of Processed and Raw Brown Rice

Brown rice, in both its raw and fermented forms, have been studied for its potential health benefits, particularly its anti-cancer and anti-inflammatory activities. The mechanisms and compounds responsible for these activities stem from the distinct nutrient and bioactive compound profile of brown rice and the products formed during its fermentation [16]. Cancer primarily results from the unchecked proliferation of abnormal cells leading to tissue damage. According to the World Health Organization (WHO), cancer ranks as the second leading cause of death globally, with projections suggesting approximately 10 million cancer-related fatalities in 2020—this accounted for one in every six deaths [11]. Consequently, substantial research into naturally occurring substances was necessary to prevent and mitigate cancer incidence. Rai and Jeyaram [24] posited that consuming fermented foods could significantly reduce the risk of various cancers and offer protection against numerous disorders. Fermented brown rice was among the foods documented to diminish cancer risk. For instance, brown rice fermented by *Aspergillus Orzae* demonstrated the potential to suppress and prevent the onset of N-methyl-N′-nitro-N-nitrosoguanidine (MNNG)-induced stomach tumors [25]. Moreover, it showed prophylactic properties against tumor development induced by 4-(methylnitrosamino)-1-(3-pyridyl)-1-butanone (NNK), a crucial agent in lung cancer, the leading cause of cancer deaths in 2020 [26]. This fermented rice has also been associated with the inhibition of tumor formation in the colon [27], liver [28], and esophagus [29].

From the early 20th century, debates persisted regarding the correlation between inflammation and cancer. Environmental agents that trigger inflammation play a pivotal role in cancer genesis. Notably, chronic inflammation had a significant association with cancer development. Mantovani et al. [30] estimated that inflammation contributed to nearly 20% of cancer-induced deaths. Brown rice fermented with *Aspergillus oryzae* has been observed to obstruct inflammatory cell infiltration at inflammatory sites without reducing bone marrow density, thereby potentially diminishing inflammation-induced cancer risks. This evidence underscores the anti-inflammatory attributes of fermented

brown rice, highlighting its potential as a natural anti-cancer resource. Furthermore, fermented brown rice with *Aspergillus oryzae* has been found to have anti-inflammatory effects in skin irritation and allergic rhinitis scenarios [31]. It was imperative to acknowledge that while these findings are promising, many were derived from in vitro experiments or animal models. Comprehensive clinical trials in humans are requisite to corroborate the exact therapeutic potential of both raw and fermented brown rice. It is essential to approach food as part of a holistic health and prevention paradigm rather than an isolated remedy.

## 8. Neuro-Protective and Anti-Stress Effects of Processed Brown Rice

Processed brown rice, especially when subjected to specific treatments such as fermentation, exhibits enhanced neuro-protective effects compared to its unprocessed counterpart. An explanation for this lay in the increased Gamma Aminobutyric Acid (GABA) content, primarily a result of fermentation. GABA plays a significant role in reducing neural excitability and conferring neuro-protective effects [20]. Elevated levels of GABA and ferulic acid are linked to improved brain function and the mitigation of neurological disorders, with their enhanced antioxidant activity significantly managing neurodegenerative diseases such as Alzheimer's and Parkinson's [18]. The reduction in anti-nutritional compounds, like phytates, and the presence of essential minerals like magnesium and zinc further contribute to neural health. While the intrinsic compounds in unprocessed brown rice already provide neuro-protection, specific processing methods can amplify these effects by increasing certain compound concentrations or introducing new beneficial elements. However, the actual impact of these effects on human health largely depends on various factors, including the processing method and an individual's overall diet and lifestyle [24].

The brain, housed within the skull, alongside the spinal cord, constitutes the central nervous system. Positioned at the central control point of our bodily systems, it performs myriad functions including muscle coordination, sensory information processing, language, learning and memory, homeostasis maintenance, and hormone secretion. Even at rest, the brain consumes about 20% of inhaled oxygen. Being highly metabolically active, it necessitates a consistent oxygen supply to maintain regular physiological functions [17]. Nonetheless, oxygen consumption produces free radicals, increasing the demand for more oxygen, which in turn escalates reactive oxygen species (ROS) production. As suggested by Shukla et al. [32], ROS might be implicated in the pathogenesis of various neurodegenerative disorders like Parkinson's disease, Alzheimer's disease, multiple sclerosis, and motor neuron disease. Phani Kumar et al. [33] posited that among the different ROS, $H_2O_2$ predominantly induces cell death via the apoptotic pathway. The research of Divate et al. [34] revealed that brown rice fermented with *Xylaria nigripes* significantly enhances neuro-protective potential, with an ethanol extract from *Xylaria nigripes* fermented brown rice offering optimum protection against H2O2-induced damage. Moreover, peptides from brown rice fermented by *Limosilactobacillus reuteri* were shown to counteract $H_2O_2$-induced cellular damage [20]. Rice bran, discarded during white rice production, differentiates brown rice from its white counterpart. Studies have demonstrated that rice bran fermented with *Lactobacillus plantarum* Hong (KFCC 11556P) can shield neurons from damage, particularly from hydrogen peroxide, and might also rejuvenate or bolster functionality [35] (Figure 6).

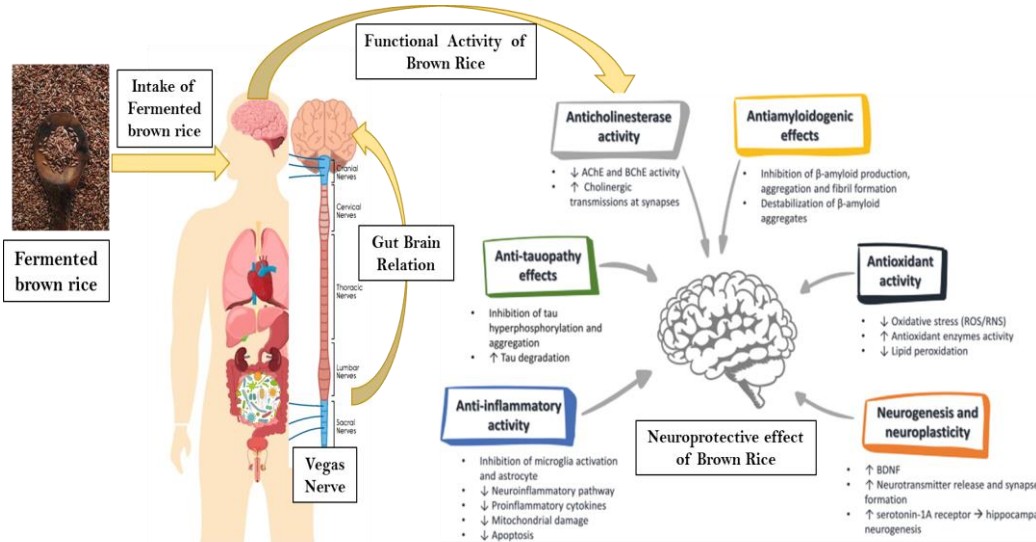

**Figure 6.** Schematic representation of the neuroprotective and anti-stress effects of brown rice, focusing specifically on its impact on neurotransmitter compounds. This diagram offers a detailed view of how brown rice contributes to brain health and stress management. It starts by identifying the key bioactive components in brown rice that are responsible for these effects, such as certain amino acids, vitamins, and minerals, known for their roles in brain function and mental well-being (different functions brain function enhanced were highlighted with different color). This comprehensive visual guide aims to showcase the multifaceted benefits of brown rice in supporting neurological health and its potential in managing stress and protecting against cognitive decline.

## 9. Enhancement of Sensory and Textural Properties of Processed Brown Rice: Consumer Preference Coupled with Improved Nutritional Properties

Rice is a staple food consumed globally. Although various cooking methods can diminish some of its nutritional value and quality, brown rice, with its bran layer, offers distinct nutritional advantages [11]. However, the presence of this bran layer can prolong cooking time and result in a coarser texture compared to white rice. These limitations reduce brown rice's appeal in the culinary domain, necessitating methods to retain its benefits while mitigating its drawbacks [17]. To address this, we studied brown rice's moisture absorption rate and cooking time, hypothesizing that a higher moisture absorption would equate to reduced cooking durations and softer textures. Subsequently, we found that fermented brown rice exhibited a superior water absorption rate than its unfermented counterpart, reducing its cooking time by 1.5 times [12]. The fermentation process induced surface cracks in the rice grains, facilitating moisture penetration during cooking and yielding a creamier texture [36].

Investigations into texture, often gauged by hardness (the force needed to deform a substance), revealed that fermented brown rice was 33.87% less hard than unfermented brown rice [12]. This suggests that fermentation alters the starch endosperm structure, increasing its permeability to water and vapor. Moreover, fermented brown rice demonstrated superior moisture absorption and elasticity compared to unfermented varieties [12]. These findings underscore the positive impacts of fermentation on the rheology and texture of brown rice [37].

## 10. Development of Food Products—Traditional Processing Technologies for Modern Application

### 10.1. Baked and Steam-Based Processed Products

Growing consumer interest in more nutritious foods has spurred many studies on the development of healthful and practical food options. Expenditure on grains, including groceries, bread, and rice, has continued to increase [12]. Consequently, more products

are now being crafted using whole grains to ensure nutritional benefits. Bread, a staple consumed worldwide, has seen an influx of gluten-free varieties made from whole grains instead of wheat flour [18]. However, gluten-free products often lack the appealing appearance of their conventional counterparts due to the absence of gluten's viscoelastic networks, resulting in bread that is less fluffy. Nevertheless, integrating fermented brown rice flour showed significant improvements in bread's volume, texture, and rheological properties [37]. Similarly, using fermented brown rice in the preparation of steamed rice cakes (at temperatures lower than those used for baking) enhanced the nutritional value, flavor, and texture of these rice cakes [38].

*10.2. Development of Healthy Snacks and Ready-to-Eat Food Products*

The extrusion process, pivotal in snack production, affects the physicochemical and physiological activities post-extrusion of fermented brown rice. Snacks produced with this rice exhibit a richer extraction of beneficial compounds [23]. Research has indicated that the phenolic compound concentration rises with an increasing screw speed [38]. Therefore, utilizing fermented brown rice in snack production could potentially lead to nutritionally superior and healthier products. Noodles, like rice and bread, are primary staple foods. Given current fast-paced lifestyles, many opt for noodles as a primary nutritional source. A study by Seo et al. [39] revealed that noodles made from fermented brown rice undergo changes in their composition. They possess slightly less crude protein and fat than their white rice counterparts, while the moisture content in the fermented brown rice is roughly double. The fermentation process also augments the pliability of the material. Sensory analysis by Seo et al. [39] suggested that fermented brown rice surpasses white rice in attributes like color, aroma, taste, texture, and overall acceptability. Thus, noodles incorporating fermented brown rice offer enhanced nutritional, textural, and sensory profiles [37]. Beverages crafted using fermented brown rice show an uptick in GABA content, a significant reduction in oxidative stress, and improved nutritional and sensory profiles, highlighting their potential as functional beverages [4,40–45].

## 11. Economic and Consumer-Oriented Environmental Benefits

Brown rice and white rice differ primarily based on the presence or absence of rice bran. Economically and environmentally, brown rice offers more benefits than white rice [24]. This is because the wheat grains are removed during the milling process to produce white rice, leading to a significant wastage of rice bran [28,31]. Moreover, fermented brown rice not only boosts nutritional value by altering rice's nutritional composition and increasing its beneficial content but also contains a substantial number of phenolic chemicals and plant polyphenols [1,7,11]. This rice variant further offers health advantages by preventing various diseases, thanks to its antioxidant [1,15], antidiabetic [21,22], anti-obesity [15], and anti-inflammatory [27,28] properties. As a result, fermented brown rice is highly valued in the food industry and is viewed as a hallmark of health and nutrition in contemporary society [18]. The market has seen a proliferation of products like bread, noodles, snacks, and soy milk made with fermented brown rice [6]. Moreover, as the demand for cosmetics primarily composed of herbal extracts and fruits rises, there is an emerging trend of developing cosmetics infused with fermented brown rice [7].

## 12. Conclusions and Future Functional Role

The nutritional superiority of brown rice (BR) and its fermented counterpart over milled rice is well documented in the literature, particularly in terms of their higher protein, fat, mineral, and vitamin contents. These rice varieties are also abundant in beneficial bioactive substances such as flavonoids, phenolic acids, γ-oryzanol, and GABA (Gamma-Aminobutyric Acid). Studies have highlighted the various health benefits associated with different rice varieties, including their antioxidant, anti-diabetic, and anti-cancer properties. Consuming bran or fermented bran is deemed healthier than choosing polished or raw rice (RR), leading to a growing advocacy for brown rice, its fermented forms, and its derivatives

as functional foods that promote health. In the food industry, innovative products like noodles, bread, and snacks have been successfully developed using both BR and fermented BR. However, the widespread commercial success of these products is hindered by challenges, such as a low sensory appeal and limited shelf life. This creates a pressing need for innovative processing techniques that can yield palatable, high-quality food products from BR and fermented BR. Such advancements would not only enhance the appeal of these healthier rice variants but also encourage consumers to prefer them over traditional milled or raw rice options. This shift towards healthier rice-based products aligns with the growing consumer interest in functional foods that contribute to overall well-being.

**Author Contributions:** Conceptualization, M.-J.L. and K.B.; formal analysis, R.C. and Y.-J.J.; funding acquisition, D.-H.O., R.C., M.-J.L. and K.B.; investigation, D.-H.O., R.C. and S.-J.P.; methodology, R.C., K.B., S.V., Y.-J.J. and S.-J.P.; formal analysis, R.C., K.B., S.V., Y.-J.J and S.-J.P., project administration, D.-H.O.; writing—original draft preparation M.-J.L. and K.B.; writing—review and editing, M.-J.L., K.B., R.C. and Y.-J.J. All authors have read and agreed to the published version of the manuscript.

**Funding:** This research was funded by (National Research Foundation of Korea (N.R.F.)) grant number [2018007551], Basic Science Research Program (NRF Grant number: 2021R1A6A1A03044242). and Brain Korea (B.K.) 21 Plus Project [Grant No. 22A20153713433].

**Acknowledgments:** This work was supported by the Korea Research Fellowship Program through the National Research Foundation of Korea (N.R.F.), funded by the Ministry of Science, in the Young Researchers Program [2018007551]; funded by Future F Biotech Co., Ltd. Brain Korea (B.K.) 21 Plus Project [Grant No. 4299990913942], funded by the Korean Government, Republic of Korea; and Basic Science Research Program (NRF Grant number: 2021R1A6A1A03044242).

**Conflicts of Interest:** The authors declare no conflict of interest.

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
