# Peer review of "Exploring the Impact of Fermentation on Brown Rice: Health Benefits and Value-Added Foods—A Comprehensive Meta-Analysis"

_fermentation, doi:10.3390/fermentation10010003_

Round 1

Reviewer 1 Report

Comments and Suggestions for Authors

Title: Oryza sativa should be italic. I suggest delete ‘A Systematic Review of’ from title.

Line 77, ‘Our review examines’ should be ‘Our review examined.

Table 1 is too draft, I suggest author revise it. Delete part Benefits and the description in table. Move the description of Benefits to text.

Figure 1. I suggest redraw Figure 1, delete the table at the bottom of the image, and draw the table as images.

Figure 2. I suggest redraw Figure 2. in this image, too many affect, you should express as inhibit or increase, etc.

Figure 3. I suggest redraw Figure 3. add the Antioxidant indicator to image.

Figure 5. I suggest redraw Figure 5. too draft.

References list: I suggest add references related to your topic. I suggest update the references, and keep the references from 2020 more than 70%.

The writing English should be improved by native English speaker. I strongly suggest to check the manuscript carefully, especially please check the grammar and the completeness of the sentences once again. And please check the tense of the sentences. There should be consistency throughout the manuscript using past tense.

Comments on the Quality of English Language

Title: Oryza sativa should be italic. I suggest delete ‘A Systematic Review of’ from title.

Line 77, ‘Our review examines’ should be ‘Our review examined.

Table 1 is too draft, I suggest author revise it. Delete part Benefits and the description in table. Move the description of Benefits to text.

Figure 1. I suggest redraw Figure 1, delete the table at the bottom of the image, and draw the table as images.

Figure 2. I suggest redraw Figure 2. in this image, too many affect, you should express as inhibit or increase, etc.

Figure 3. I suggest redraw Figure 3. add the Antioxidant indicator to image.

Figure 5. I suggest redraw Figure 5. too draft.

References list: I suggest add references related to your topic. I suggest update the references, and keep the references from 2020 more than 70%.

The writing English should be improved by native English speaker. I strongly suggest to check the manuscript carefully, especially please check the grammar and the completeness of the sentences once again. And please check the tense of the sentences. There should be consistency throughout the manuscript using past tense.

Reviewer 2 Report

Comments and Suggestions for Authors

The article has summarized the henefits of whole brown rice, and the quotation is full.

But there are some issues to consider:

Table 1 should be readjusted.

Table 1, Redrice, the total content of protein, carbohydrate, fat and ash is more than 100%. In principle, the inorganic and organic content is 100%. Please verify.

Line 141, 231, etc., Casematters.

Comments on the Quality of English Language

Minor revision.

Reviewer 3 Report

Comments and Suggestions for Authors

Dear Authors of manuscript entitled „A Systematic review of fermentation’s effects on oryza sativa (brown rice): a meta-analysis of health benefits and value-added food”, please find my comments and suggestions related with your work:

Please add to the abstract with the most important conclusions that emerge after the analysis described in the article.

45-47 Giving the study's title is clumsy at this point. Please provide data, facts, results relevant to the article and cite the literature source, as is done with a scientific publication.

48-56, This fragment, even for an introduction, is too general, too few details and examples are provided. Please provide examples, including: what kind of various bioactive compounds?, what kind of probiotic properties? what kind of health benefits? What species of harmful microorganisms, and so on…

All Introduction section is written too generally, the examples and explanations have to be done in this part.

L62-63, please explain what are the environmental risks after using this method?, Please give more detailes: Fermentation of brown rice and probiotics yields bioactive chemicals that enhance health. Globally, fermented foods, enriched in flavor, aroma, and nutrition, have bacteria and metabolites that significantly impact human health. – wtah is “fermentation of probiotics”? I don’t understand this term, please add some examples of this “impact of human health” etc.

What I miss in this article is specific data and specific information. There are a lot of general descriptions, too general if the article is to be a meta-analysis.

What I read is rather general knowledge, something like a popular science review study.

Meta-analysis requires the comparison of specific data from many publications from the same research field. It requires a statistical analysis of the results, a deep analysis, a meta-analysis is a separate examination of the available data, and not a quotation of selected fragments of other publications.

Why didn't the authors explain what exactly fermentation is? How does it proceed? What is its mechanism of fermentation? What microorganisms are involved in it?

After reading 131 lines of the article, I still do not see any name of the microorganisms that may be involved in fermentation.

Table 1 absolutely needs to be modified, it cannot be accepted in the current version and requires deep editing. Additionally, please provide additional information about the benefits of eating white rice, as there are some. If there are any disadvantages (white and brown rice), they should also be put in this table.

The manuscript lacks mentioned to the included figures.

Overall, the article provides a lot of interesting information about the benefits of eating brown rice. In my opinion, however, the article cannot necessarily be called a "meta-analysis", I would rather suggest it is a "review". Before an article is considered for publication, it must be corrected and supplemented, tables must be modified, and figures presented in the work must be cited.

Round 2

Reviewer 1 Report

Comments and Suggestions for Authors

The manuscript was improved enough, I suggest accept it.